# Optimization of Alpha-Amylase Production by a Local *Bacillus paramycoides* Isolate and Immobilization on Chitosan-Loaded Barium Ferrite Nanoparticles

**Merehan Hallol** [1,†], **Omneya Helmy** [2,*,†] (ID), **Alla-Eldien Shawky** [2], **Ahmed El-Batal** [1] and **Mohamed Ramadan** [2]

[1] Department of Drug Radiation Research, National Center for Radiation Research and Technology, Egyptian Atomic Energy Authority, Cairo 11562, Egypt; merehan.hallol87@gmail.com (M.H.); aelbatal2000@gmail.com (A.E.-B.)

[2] Department of Microbiology & Immunology, Faculty of Pharmacy, Cairo University, Cairo 11562, Egypt; shawkyalaa608@gmail.com (A.-E.S.); mohamed.abdelhalim@pharma.cu.edu.eg (M.R.)

\* Correspondence: omnia.helmy@pharma.cu.edu.eg

† These authors contributed equally to this work.

**Abstract:** We set out to isolate alpha-amylase producers from soil samples, optimize the production, and immobilize the enzyme on chitosan-loaded barium ferrite nanoparticles (CLBFNPs). Alpha-amylase producers were isolated on starch agar plates and confirmed by dinitrosalicylic acid assay. The potent isolate was identified by phenotypic methods, 16S-rRNA sequencing, and phylogenetic mapping. Sequential optimization of α-amylase production involved the use of Plackett–Burman (P–BD) and central composite designs (CCD), in addition to exposing the culture to different doses of gamma irradiation. Alpha-amylase was immobilized on CLBFNPs, and the nanocomposite was characterized by X-ray diffraction, Fourier-transform infrared spectroscopy, and scanning electron microscopy, with energy-dispersive analysis of X-ray analysis. Forty-five α-amylase producers were isolated from 100 soil samples. The highest activity (177.12 ± 6.12 U/mg) was detected in the MS009 isolate, which was identified as *Bacillus paramycoides*. The activity increased to 222.3 ± 5.07 U/mg when using the optimal culture conditions identified by P–BD and CCD, and to 319.45 ± 4.91 U/mg after exposing the culture to 6 kGy. Immobilization of α-amylase on CLBFNPs resulted in higher activity (246.85 ± 6.76 U/mg) compared to free α-amylase (222.254 ± 4.89 U/mg), in addition to retaining activity for up to five cycles of usage. Gamma irradiation improved α-amylase production, while immobilization on CLBFNPs enhanced activity, facilitated enzyme recovery, and enabled its repetitive use.

**Keywords:** alpha-amylase; gamma irradiation; immobilization; *Bacillus paramycoides*; chitosan; barium ferrite; nanoparticles

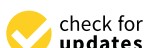



## 1. Introduction

Amylases seize around 25% of the overall industrial enzyme market. They have many applications in the food, fermentation, pharmaceutical, detergent, textile, and paper industries [1]. Marketed alpha-amylases are produced from plants, animals, and microorganisms, but their production from the first two sources is relatively lower [2,3]. This is because of the rapid growth of microorganisms, which speeds up the production of enzymes. Microorganisms can be easily manipulated using genetic engineering and tailored to cater to the needs of growing industries, such as obtaining enzymes with desired characteristics, including thermostability; moreover, they are easy to handle, require lesser space, and serve as more cost-effective sources [4]. Most of the industrial enzymes are produced by either *Bacillus* or *Aspergillus* species, of which some can be produced in a single fermentation medium, resulting in a cost-effective production process and improved enzymatic stability [5]. Microbial alpha-amylase producers from soil have previously been reported in the literature [4,6].

Designing an effective fermentation medium for maximal enzyme production is a critical process that affects the product yield [7–10]. Response surface methodology (RSM) is a group of statistical techniques used for designing experiments, building models, evaluating the effects of different factors, and determining the optimal values for the needed response [11]. Statistical methods, such as Plackett–Burman (P–B) design, have been used to optimize the production of several industrial enzymes, including amylases, proteases, catalase, and chitinase [12], thus decreasing the time and effort while enhancing the yield [13].

Means to improve the yield of the desired enzymes by wild-type strains involve exposing them to physical and chemical mutagens, such as UV light, γ-rays, and antibiotics, which can induce changes in their microbial genomes [14]. Gamma rays emitted from the disintegration of Co60 radioisotopes are among the most commonly used physical mutagens in practice [15,16]. They induce the formation of reactive oxygen species (ROS) in the irradiated cells, which interact with DNA and RNA, resulting in nucleic acid mutation, or even cell death [15,16]. Sometimes, they can create useful mutations at specified locations in a genome [17]. Increases in the productivity and activity of many enzymes—such as alpha-amylase, dehydrogenases, phosphatase, uricase and peroxidase—have been detected after exposing the microbial producer to γ-irradiation [14,16,18].

The use of soluble enzymes in industrial applications is limited, because of their non-reusability, instability, and denaturation, resulting in higher costs for the process [19]. Enzyme immobilization allows the simple separation of the enzyme from its reaction medium, and multiple reusability, hence reducing the cost of the process [20,21]. Chitosan is used for enzyme immobilization owing to its inert nature, biocompatibility, low cost, hydrophilic nature, biodegradability, and non-toxicity [19,22]. This can occur either via the entrapment of the enzyme in the chitosan beads, or by covalent binding to the amino groups in chitosan films [23,24]. Many industrial enzymes are immobilized on chitosan, including amylases, proteases, catalase, and chitinase [25]. Due to their high surface area and improved dispersibility in liquid media, nanoparticles (NPs) have become the most commonly used support materials for many enzymes [26]. Magnetic nanoparticles (MNPs), with the additional advantage of easy separation from the reaction mixture using an external magnet, have gained attention in the past few years; they have numerous applications in biomedicine, biotechnology, and biochemistry [26–28].

MNPs can be prepared from inexpensive starting materials that can be adjusted by proper surface modification, and are used in enzyme immobilization owing to their biocompatibility, small size, low toxicity, superparamagnetism, and easy separation [26–28]. Among the various magnetic nanoparticles, $Fe_3O_4$ nanoparticles are frequently used as the core magnetic support, due to their ease of synthesis, low cost, and considerably high magnetic susceptibility [26]. Using MNPs in industry is widely accepted, as they provide a high surface area, which results in a high binding efficiency, increases the reaction rate and turnover number, and decreases mass transfer resistance and contamination [29,30]. This enables their repetitive use at a low cost [31]. Superparamagnetic iron oxide (magnetite and maghemite) nanoparticles are in use because of their improved biosubstance binding and their potential in targeting and delivery systems. They have a high surface-to-volume ratio and, hence, higher surface energy, with excellent magnetic properties compared to larger particles [32]; moreover, they possess an additional characteristic compared to other conventional support materials—selective and secure enzyme recovery from the medium under magnetic force. Hence, there is no need for expensive liquid chromatography systems, centrifuges, filters, or other equipment [33]. The functionalization of the MNPs involves the creation of functional groups on their surface to bind the desired moiety and improve the dispersion properties. Various organic and inorganic coating materials are used for the functionalization of the MNPs [28,34–36].

The isolation of microorganisms from extreme conditions or contaminated sites offers microorganisms with unusual properties and activities. The Egyptian soil is an underexplored source of enzyme producers [18]. We set out to isolate alpha-amylase producers from

Egyptian soil, and optimize their production using statistical methods and gamma irradiation, along with immobilization of the produced alpha-amylase on chitosan-loaded barium ferrite nanoparticles and evaluation of the activity and stability of the prepared composite.

## 2. Materials and Methods

### 2.1. Collection of Soil Samples and Isolation of Alpha-Amylase Producers

We collected soil samples (*n* = 100) from different agricultural fields—including broccoli (*Brassica oleracea*, *n* = 13), peanut (*Arachis hypogaea*, *n* = 12), rice (*Oryza sativa*, *n* = 15), cassava (*Manihot esculenta*, *n* = 10), capsicum (*Capsicum annuum*, *n* = 19), eggplant (*Solanum melongena*, *n* = 16), soya bean (*Glycine max*, *n* = 18), maize (*Zea mays*, *n* = 19), beet palm (*Beta vulgaris*, *n* = 11), orange (*Citrus aurantium*, *n* = 12), tangerine (*Citrus reticulate*, *n* = 10), apricot (*Prunus armeniaca*, *n* = 15), pomegranate (*Punica granatum*, *n* = 20), fig (*Ficus carica*, *n* = 8), apple (*Malus domestica*, *n* = 12), guava (*Psidium guajava*, *n* = 9), pear (*Pyrus communis* L., *n* = 11), grapes (*Vitis vinifera* L., *n* = 7), and olives (*Olea europaea*, *n* = 13)—from October 2017 to December 2018. Recovery of alpha-amylase producers from soil samples was performed according to the method of Singh et al. Briefly, 5 g of soil sample was inoculated in a flask containing 100 mL of mineral enrichment broth medium (MEB, Sigma-Aldrich, Burlington, MA, USA) and incubated with shaking at 150 rpm for 24 h at 37 °C [6]. We isolated cultures on starch agar plates to recover amylolytic microorganisms. Potential alpha-amylase producers were further sub-cultured on mineral enrichment medium (MEM), and pure isolates were maintained on nutrient agar plates [3,37].

### 2.2. Dinitrosalicylic Acid Assay (DNS) to Detect Alpha-Amylase Activity

Suspected isolates were inoculated in mineral enrichment broth (MEB) and incubated at 37 °C with shaking at 100 rpm; samples were taken every 24 h and filtered using ash-free filter paper. The cell-free filtrate was collected and used as the source for crude amylase. The protein content of the cell-free filtrate was quantified using the Bradford protein assay [38]. The DNS assay was performed according to the method of Padhiar and Kommu [39]. The reaction mixture comprised 0.5 mL of crude enzyme and 0.5 mL of 1% soluble starch (Millipore Sigma-Aldrich, Burlington, MA, USA), dissolved in 0.1 M phosphate buffer at pH 7. The reaction was incubated at 50 °C for 3 min, and stopped by the addition of 1 mL of 3,5-dinitro salicylic acid (Millipore Sigma-Aldrich, Burlington, MA, USA) and boiling for 5 min. The amount of released reducing sugar (glucose) was measured at a wavelength of 540 nm using a spectrometer (UV–Visible spectrometer T60 Vis, Leicestershire, UK), where one unit was defined as 1 µmol glucose equivalent per min. The alpha-amylase activity was calculated according to the following equation: activity ($Umg^{-1}$) = released glucose (µmol)/(amount of $\alpha$-amylase (mg) $\times$ min).

### 2.3. Identification of the Most Potent Alpha-Amylase Producer MS009

Pure cultures were identified by their colony morphology, Gram-staining, microscopic examination, and biochemical characteristics [40]. Genomic DNA was extracted from colonies grown on nutrient agar for 18 h using GeneJet genomic DNA purification kit (Thermo Fisher Scientific, Vilnius, Lithuania). PCR amplification of 16S rRNA genes was performed in a final reaction volume of 50 µL, using the following 16S rRNA universal primers: 63F 5′-CAGGCCTAACACATGCAAGTC-3′ and 1387 R 5′-CGGCGGWGTGTACAAGGC-3′, at a concentration of 20 µM each [41], along with 5 µL of template DNA in 1× Maxima Hot Start PCR Master Mix (Thermo Fisher Scientific, Lithuania, EU), using a Techne thermal cycler (Cole-Parmer, Chicago, IL, USA). The cycling parameters were denaturation at 95 °C for 10 min; 35 cycles each of 95 °C for 30 s, 65 °C for 1 min 30 s, and 72 °C for 1 min; and a final extension at 72 °C for 10 min. The amplified product was purified using the Gene JET™ PCR Purification Kit (Thermo Thermo Fisher Scientific, Lithuania, EU), and sequenced using the ABI3730XL sequencer (Macrogen, Seoul, Korea). The obtained sequence was blasted against the nucleotide database using the BLASTn tool of the US National Center for Biotechnology Information (NCBI). Homology search was also carried out using the basic

BLASTn tool of the US National Center for Biotechnology Information (NCBI). Phylogenetic analysis was performed using the neighbor-joining methods and minimum evolution tree, with bootstrap values based on 1000 replications, and the phylogenetic tree was constructed with the MEGA package version 11, according to the Jukes–Cantor method [34,42,43].

### 2.4. Optimization of Alpha-Amylase Production

Submerged fermentation in MEB was performed to study the effects of various physiochemical parameters on the production of α-amylase by MS009. Different parameters, selected from the literature, were tested to select the most significant factors affecting enzyme production, including the use of starch, peptone, yeast extract, Tween 80, and $CaCl_2$ in culture medium, along with adjusting the pH of the culture medium, aeration (culture-to-flask volume), inoculum size, shaking speed, incubation period, and temperature [44]. Optimization of the fermentation medium, for maximal enzyme production, was achieved using Plackett–Burman design (P–BD) as an initial optimization step, followed by central composite design (CCD), based on RSM, as a second phase [45]. We used the Design-Expert 7.0 (Stat Ease Inc., Minneapolis, MN, USA) statistical software to design the experiment, in regression analysis of the experimental data, and in plotting the relationships between variables.

Fermentation was carried out in 250 mL Erlenmeyer flasks, containing 100 mL of sterile culture media. Alpha-amylase production was monitored in MEB fermentation medium under different culture conditions using P–BD in 12 independent reactions generated by Design-Expert 7.0 software. The tested variables (in two-level forms) included temperature (35 °C or 40 °C), shaking speed (150 rpm or 180 rpm), pH (7 or 8), inoculum size (2.5 mL or 5 mL; the bacterium was originally grown in MEB medium to an $OD_{600}$ of 0.3), culture volume/total flask volume mL (1/5 or 1/2.5), incubation period (48 h or 72 h), starch (0.5% or 1%) (Sigma-Aldrich, Burlington, MA, USA), peptone (0.3% or 0.6%) (Sigma-Aldrich, Burlington, MA, USA), yeast extract (0.3% or 0.6%) (Sigma-Aldrich, Burlington, MA, USA), Tween 80 (0.1% (*v/v*) or 0.2% (*v/v*)) (Sigma-Aldrich, Burlington, MA, USA), and calcium chloride (1% or 2%) (Sigma-Aldrich, Burlington, MA, USA) (Table 1). Alpha-amylase production was assayed via the DNS method and scored as U/mg. Every experiment was repeated three times independently, and the obtained result was the average. The significant variables obtained from P–BD were further subjected to a second phase of analysis by CCD based on response surface methodology (RSM) (Table 2).

**Table 1.** Independent variables affecting the production of alpha-amylase by the MS009 isolate, and their levels in the Placket–Burman design.

| Tested Variables | Level 1 | Level 2 |
|---|---|---|
| Starch g% | 0.5 | 1 |
| Peptone g% | 0.3 | 0.6 |
| Yeast extract g% | 0.3 | 0.6 |
| Culture volume/total flask volume mL | 1/2.5 | 1/5.0 |
| Tween 80% (*v/v*) | 0.1 | 0.2 |
| $CaCl_2$ g% | 1 | 2 |
| Incubation period (h) | 48 | 72 |
| Incubation temperature (°C) | 35 | 40 |
| Shaking speed (rpm) | 150 | 180 |
| pH | 7 | 8 |
| Inoculum size mL * | 2.5 | 5 |

* The bacterium was originally grown in MEB medium to an $OD_{600}$ of 0.3.

**Table 2.** Variables affecting the production of alpha-amylase by the MS009 isolate, and their levels in the central composite design.

| Variable | Peptone (g%) | Culture Volume/250 mL Flask | CaCl$_2$ (g%) | Incubation Period (h) |
|---|---|---|---|---|
| Level 1 | 0.15 | 25 | 1 | 48 |
| Level 2 | 0.45 | 50 | 1.5 | 60 |
| Level 3 | 0.3 | 75 | 2 | 72 |
| Level 4 | 0.6 | 100 | 2.5 | 84 |

*2.5. Effect of Gamma Irradiation on Alpha-Amylase Production*

Gamma irradiation was applied within a gamma-irradiation system using Co-60 Gamma (Gamma cell 4000-A-India), at a dose rate of 1.429 kGy/h at the time of the experiment, at the National Center for Radiation Research and Technology (NCRRT). We irradiated freshly cultured MS009 slants with different doses of gamma irradiation (1, 2, 4, 6, 8, and 10 kGy), followed by sub-culturing in 50 mL of MEB and incubation at 37 °C for 48 h. Then, 2.5 mL of the gamma-irradiated MS009 isolate culture (O.D. 600 = 1) was incubated under the optimal reaction conditions, identified previously. A non-irradiated MS009 slant was used as a negative control [46]. The enzymatic activity was determined by the DNS method, as mentioned earlier.

*2.6. Partial Purification of Alpha-Amylase*

MS009 cultures were centrifuged at 10,000 rpm for 10 min at 4 °C (Hettich Universal 16R, Tuttlingen, Germany). The extracellular crude enzyme, in the cell-free cultures (60 mL), was partially purified by precipitation with either various concentrations of ammonium sulfate (NH$_4$)$_2$SO$_4$ (40%, 50%, 60%, 70%, and 80%) to attain 60% saturation, or acetone in a ratio of 1:2 saturation, and kept at 4 °C for 24 h. The precipitate was collected by centrifugation at 10,000 rpm for 10 min at 4 °C, and dissolved in 60 mL of 100 mM phosphate buffer (pH 7.0); this was followed by overnight dialysis against phosphate buffer (10 mM, pH 7.0) at 4 °C, and the dialysate was stored at −20 °C [47]. The protein content of the partially purified alpha-amylase solution was quantified using the Bradford protein assay [38], and the purification factor was calculated [47]. Each experiment was repeated three times independently, and the obtained result was the average.

*2.7. Immobilization of the Partially Purified Alpha-Amylase on Chitosan-Loaded Barium Ferrite Nanoparticles (CLBFNPs)*

Barium ferrite magnetic nanoparticles (BFMNPs) were prepared according to the method of El-Batal et al. [33]. Chitosan solution was prepared by dissolving 25 mg of chitosan in 25 mL of distilled water via stirring. BFMNPs were loaded with chitosan by adding 25 mg of BFMNPs to the prepared chitosan solution, followed by gentle shaking at 25 °C for 2 h. Chitosan-loaded barium ferrite nanoparticles (CLBFNPs) were washed with distilled water and recovered by magnetic decantation [33,48]. Immobilization of α-amylase on CLBFNPs was performed by gentle shaking of a mixture of α-amylase (1 mg/mL) in PBS (pH 7.0) and CLBFNPs (1 mg/mL) at a ratio of 1:1 for 2 h. Alpha-amylase immobilized on CLBFNPs was magnetically separated from the solution and washed three times with PBS (pH 7.0) [48]. The alpha-amylase remaining in the original solution and in the washed solutions was measured using the Bradford protein assay to estimate the amount of unbound enzyme, and the amount of immobilized enzyme per weight was calculated [38]. The activity of alpha-amylase-CLBFNPs was determined via the DNS method [49].

### 2.8. Characterization of the Prepared α-Amylase-CLBFNPs

The prepared α-amylase-CLBFNPs were characterized by X-ray diffraction (XRD) and compared to chitosan, barium ferrite, and alpha-amylase; the crystal composition and the average crystal size of the synthesized chitosan, barium ferrite, and alpha-amylase nanoparticles were determined using an XRD 6000 (Shimadzu Corporation, Tokyo, Japan) [50]. Fourier-transform infrared (FTIR) spectra were determined for alpha-amylase enzyme, barium ferrite, chitosan, and α-amylase-CLBFNPs solutions using JASCO FT-IR-300-3600 (JASCO, Mary's Court Easton, MD, USA) with applied KBr pellets techniques [33]. The surface morphology of α-amylase-CLBFNPs was examined by scanning electron microscopy with energy-dispersive analysis of X-ray (SEM/EDAX) analysis ZEISS-EVO-MA10 ( ZEISS, Oberkochen, Germany), according to the manufacturer's instructions. SEM/EDAX was used to determine the elemental form and the mapping technique to provide complete information regarding the purity and the association of α-amylase-CLBFNPs [51].

### 2.9. Effects of Different Working Temperatures and pH on the Activity of the Prepared α-Amylase-CLBFNPs

The activity of both the free and immobilized α-amylase was determined by the DNS method, as mentioned earlier. The assay was performed by incubating at different working temperatures (20, 30, 40, 50, 60, 70, 80, 90, and 100 °C) and using 0.1 M PBS with different pH values (3, 4, 5, 6, 7, 8, 9, and 10), adjusted using 0.1N HCl and 0.1N NaOH [39,52]. Each experiment was repeated three times independently, and the obtained result was the average.

### 2.10. Reusability of the Prepared α-Amylase-CLBFNPs

We used a permanent magnet for separation of α-amylase-CLBFNPs from the original enzymatic reaction, used for detection of α-amylase activity via the DNS method, for their reuse in the next enzymatic activity assay. The enzymatic activity of the residual reaction mixture was determined using the DNS method [39]. The activity of the alpha-amylase-BFCMNPs at the first cycle was considered the control. The same procedure was repeated five times, and the enzymatic activity was determined following each cycle [48,49,51]. Each experiment was repeated three times independently, and the obtained result was the average.

### 2.11. Statistical Analysis

The analysis of data was carried out using one-way analysis of variance (ANOVA,) followed by Duncan's multiple range test, using the Statistical Package for the Social Sciences (SPSS, IBM SPSS Statistics for Windows, Version 22.0. Armonk, NY, USA) ). Means were compared using the least significant difference (LSD at the 5% level) and Duncan's multiple range test at the significance $p = 0.05$.

## 3. Results

### 3.1. Isolation of Alpha-Amylase Producers from Soil

We recovered 45 alpha-amylase producers from soil samples, of which 23 were bacterial and 22 were fungal isolates (Table 3). Isolates showing a clear zone with a diameter greater than 15 mm in the starch iodine test ($n = 17$) were further confirmed for alpha-amylase production by the DNS method. A maximum specific activity of 177.12 U/mg was detected in isolate MS009, recovered from a guava field, and this isolate was used in further experiments (Table 3).

**Table 3.** Diameter of the clear zone produced by potential alpha-amylase producers in the starch iodine test, and the alpha-amylase-specific activity detected by the DNS method.

| Isolate | Source | Classification | Diameter of the Clear Zone (mm) | α Amylase Specific Activity (U/mg) |
|---|---|---|---|---|
| MA003 | AEAG | Bacteria | 8 | ND |
| MA011 | AEAG | Fungi | 8 | ND |
| MA016 | AEAG | Bacteria | 17 | ND |
| MA023 | AEAG | Fungi | 1 | ND |
| MA035 | AEAG | Bacteria | 28 | 19.22 ± 0.94 |
| MA040 | AEAG | Fungi | 12 | ND |
| MA054 | AEAG | Bacteria | 58 | 147.32 ± 5.32 |
| MA063 | AEAG | Fungi | 12 | 12.23 ± 0.67 |
| MS001 | ESFG | Bacteria | 50 | 98.99 ± 3.45 |
| MS009 | ESFG | Bacteria | 60 | 177.12 ± 6.12 |
| MS011 | ESFG | Fungi | 39 | 50.08 ± 2.05 |
| MS017 | ESFG | Fungi | 33 | 28.43 ± 1.34 |
| MS020 | ESFG | Bacteria | 5 | ND |
| MS023 | ESFG | Fungi | 45 | ND |
| MS029 | ESFG | Bacteria | 31 | 25.04 ± 1.09 |
| MS032 | ESFG | Fungi | 27 | 18.63 ± 0.88 |
| MS038 | ESFG | Bacteria | 25 | 14.56 ± 0.89 |
| MS043 | ESFG | Fungi | 58 | 140.08 ± 2.98 |
| MS057 | ESFG | Bacteria | 29 | 20.36 ± 1.02 |
| MS061 | ESFG | Fungi | 10 | ND |
| MS070 | ESFG | Bacteria | 20 | 136.862 ± 3.67 |
| MS076 | ESFG | Fungi | 30 | 21.47 ± 0.98 |
| MS079 | ESFG | Fungi | 4 | ND |
| MS083 | ESFG | Bacteria | 15 | ND |
| MS086 | ESFG | Fungi | 5 | ND |
| MS089 | ESFG | Bacteria | 5 | ND |
| MS091 | ESFG | Fungi | 10 | ND |
| MS095 | ESFG | Bacteria | 1 | ND |
| MM003 | EMVG | Fungi | 8 | ND |
| MM007 | EMVG | Bacteria | 10 | ND |
| MM012 | EMVG | Fungi | 3 | ND |
| MM017 | EMVG | Bacteria | 15 | ND |
| MM029 | EMVG | Fungi | 8 | ND |
| MM033 | EMVG | Bacteria | 45 | 9.12 ± 0.04 |
| MM036 | EMVG | Fungi | 5 | ND |
| MM040 | EMVG | Bacteria | 15 | ND |
| MM043 | EMVG | Fungi | 3 | ND |
| MM050 | EMVG | Bacteria | 10 | ND |
| MM054 | EMVG | Fungi | 2 | ND |
| MM061 | EMVG | Bacteria | 2 | ND |
| MM066 | EMVG | Fungi | 5 | ND |
| MM070 | EMVG | Bacteria | 56 | 133.11 ± 3.54 |
| MM075 | EMVG | Fungi | 10 | ND |
| MM084 | EMVG | Bacteria | 24 | 13.39 ± 0.87 |

ND: not determined. AEAG: Atomic Energy Authority gardens. ESFG: El Sharqya fruit gardens. EMVG: El Minya vegetable gardens.

### 3.2. Identification of MS009

MS009 colonies were round and creamy-white, with a smooth surface. Microscopic examination of fresh cultures revealed rod-shaped motile cells an average of 6–10 μm long

and 0.5 μm wide, and showed a Gram-positive reaction. The obtained 16SrRNA sequence of MS009 was blasted against the nucleotide database using the BLASTn tool of the US National Center for Biotechnology Information (NCBI), and it showed 97.28% identity to *Bacillus paramycoides* (MH665578.1). We deposited it in GenBank under GenBank accession no. ON024310. We constructed a phylogenetic tree based on the 16S rRNA sequence of MS009 and closely related species using MEGA 11 (Figure 1), and this further confirmed our identification.

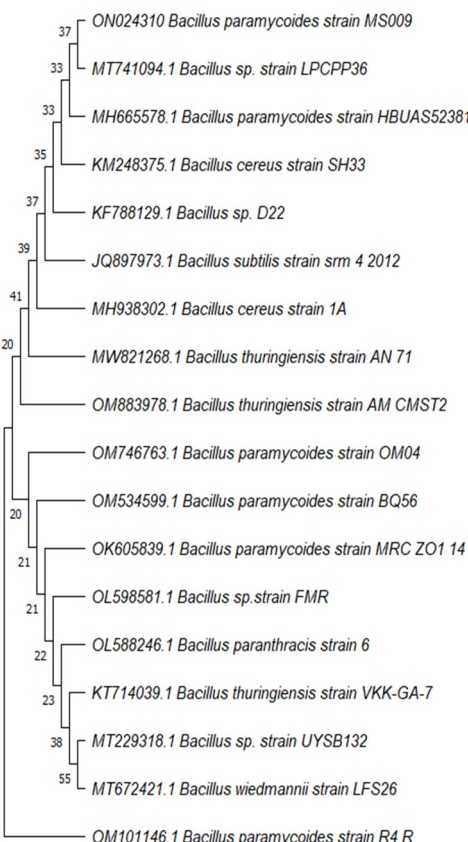

**Figure 1.** Phylogenetic tree comparing the MS009 isolate to the closely related *Bacillus* species. This was constructed using the MEGA package, version 11, based on 16S rDNA using the neighbor-joining and minimum evolution tree methods, with bootstrap values based on 1000 replications. The MS009 partial 16S rDNA was deposited in GenBank under accession no. ON024310.

*3.3. Plackett–Burman Design to Optimize Alpha-Amylase Production*

We carried out 12 experiments using a combination of different variables identified from the literature (Table 4). An optimal enzymatic activity of 222.254 ± 5.07 U/mg was recorded using the following parameters: 1 g% starch, 0.6 g% peptone, 0.6 g% yeast extract, 1/5 culture volume/total flask volume, 0.1 g% Tween, 1 g% CaCl₂, an inoculum size of 2.5 mL, pH 8, and incubation at 35 °C for 72 h, with shaking at 180 rpm. Statistical analysis of the P–BD revealed a model F-value of 20.17, which implies the significance of the model, with $p < 0.05$ indicating that the model terms are significant. The value of the predicted determination coefficient ($R^2_{Pred}$) was 0.7715, and this is comparable to the adjusted determination coefficient ($R^2_{Adj}$) (0.9127). The precision value (Adeq), which is a measure of the signal-to-noise ratio, equaled 13.001, and this indicates an adequate signal.

**Table 4.** Results of the P–BD to evaluate factors affecting alpha-amylase production by the MS009 isolate.

| Run | Starch (g%) | Peptone (g%) | Yeast Extract (g%) | Culture Volume/Total Flask Volume (mL) | Tween 80 % (v/v) | CaCl₂ (g%) | Incubation Period (h) | Temperature (°C) | Shaking (rpm) | pH | Inoculum Size (mL) | Amylase-Specific Activity (U/mg) |
|---|---|---|---|---|---|---|---|---|---|---|---|---|
| 1 | 1 | 0.6 | 0.3 | 1/2.5 | 0.2 | 2 | 48 | 35 | 150 | 8 | 2.5 | 10.616 ± 0.54 |
| 2 | 0.5 | 0.3 | 0.3 | 1/5.0 | 0.1 | 1 | 48 | 35 | 150 | 7 | 2.5 | 35.93 ± 1.32 |
| 3 | 0.5 | 0.3 | 0.6 | 1/5.0 | 0.2 | 2 | 48 | 40 | 180 | 8 | 2.5 | 25.586 ± 0.98 |
| 4 | 0.5 | 0.6 | 0.6 | 1/2.5 | 0.1 | 1 | 48 | 40 | 150 | 8 | 5 | 153.795 ± 3.03 |
| 5 | 1 | 0.3 | 0.6 | 1/2.5 | 0.1 | 2 | 72 | 40 | 150 | 7 | 2.5 | 53.03 ± 1.78 |
| 6 | 1 | 0.6 | 0.3 | 1/5.0 | 0.1 | 2 | 48 | 40 | 180 | 7 | 5 | 52.126 ± 1.56 |
| 7 | 0.5 | 0.6 | 0.3 | 1/5.0 | 0.2 | 2 | 72 | 35 | 150 | 7 | 5 | 187.685 ± 4.69 |
| 8 | 0.5 | 0.6 | 0.3 | 1/2.5 | 0.2 | 1 | 72 | 40 | 180 | 7 | 2.5 | 190.27 ± 4.69 |
| 9 | 0.5 | 0.3 | 0.3 | 1/2.5 | 0.1 | 2 | 72 | 35 | 180 | 8 | 5 | 26.812 ± 0.98 |
| 10 | 1 | 0.3 | 0.3 | 1/5.0 | 0.2 | 1 | 72 | 40 | 150 | 8 | 5 | 186.186 ± 4.78 |
| 11 | 1 | 0.6 | 0.6 | 1/5.0 | 0.1 | 1 | 72 | 35 | 180 | 8 | 2.5 | 222.254 ± 5.07 |
| 12 | 1 | 0.3 | 0.6 | 1/2.5 | 0.2 | 1 | 48 | 35 | 180 | 7 | 5 | 33.072 ± 1.69 |

Based on the estimated parameters and applying multiple regression analysis to the experimental data, the tested variables and the response variables were related by the following model equation:

$$\alpha\text{-amylase activity} = +1468.71 + 573.16 \times B + 213.85 \times C - 305.72 \times D - 585.07 \times F + 690.89 \times G + 177.78 \times H$$

B = Peptone, C = Yeast extract, D = Aeration, F = CaCl₂, G = Incubation period, H = Temp.

The effect of individual parameters studied in P–BD design is presented as a Pareto chart (Figure 2) that depicts the order of significance of the variables involved in alpha-amylase production. Peptone, aeration, CaCl₂, and incubation period are significant model terms, as values of "Prob > F" less than 0.05 indicate significance.

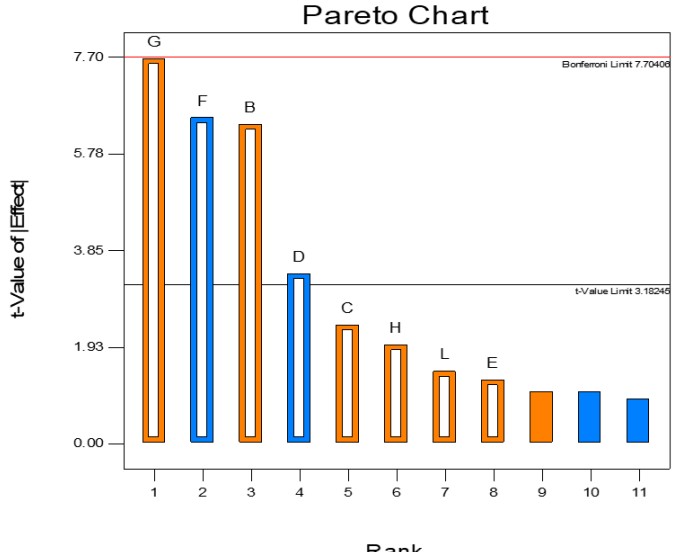

**Figure 2.** Pareto chart of Plackett–Burman design (P–BD) results showing the effects of individual factors on enzyme production, where B = peptone *, C = yeast extract, D = aeration *, E = Tween, F = CaCl₂ *, G = incubation period *, H = temperature, and L = inoculum size.* Refers to significant factors with SD 361.6.4 and R-squared 0.9603.

### 3.4. Central Composite Design to Optimize Alpha-Amylase Production

We used CCD for choosing the optimal levels of the significant variables recognized by the P–BD experiment results. The results of CCD runs ($n = 30$) for the four tested variables (peptone, aeration, CaCl$_2$, and incubation period) are recorded in Table 5. The alpha-amylase-specific activity ranged from $10.384 \pm 0.82$ to $232.456 \pm 5.98$ U/mg (Table 5). Maximal enzyme production was detected by using a high concentration of peptone 0.45 g%, culture-to-flask volume ratio 75:250 mL, CaCl$_2$ 1.5 g%, and an incubation period of 84 h. This suggests that these variables play a significant role in alpha-amylase production.

**Table 5.** Results of CCD design to optimize the variables involved in alpha-amylase production by MS009.

| Run | Peptone (g%) | Aeration (Culture Volume/250 mL Flask) (mL) | CaCl$_2$ (g%) | Incubation Period (h) | Amylase-Specific Activity (U/mg) |
|---|---|---|---|---|---|
| 1 | 0.6 | 100 | 2 | 48 | $124.72 \pm 3.01$ |
| 2 | 0.6 | 50 | 2 | 48 | $90.427 \pm 2.35$ |
| 3 | 0.3 | 100 | 2 | 72 | $225.79 \pm 5.08$ |
| 4 | 0.6 | 50 | 2 | 72 | $221.548 \pm 4.96$ |
| 5 | 0.45 | 25 | 1.5 | 60 | $108.137 \pm 2.75$ |
| 6 | 0.15 | 75 | 1.5 | 60 | $166.74 \pm 3.58$ |
| 7 | 0.3 | 50 | 2 | 48 | $128.84 \pm 3.26$ |
| 8 | 0.3 | 100 | 1 | 48 | $81.36 \pm 2.43$ |
| 9 | 0.3 | 50 | 1 | 48 | $112.693 \pm 2.97$ |
| 10 | 0.6 | 100 | 1 | 48 | $118.691 \pm 2.47$ |
| 11 | 0.45 | 75 | 1.5 | 60 | $151.833 \pm 3.48$ |
| 12 | 0.45 | 75 | 1.5 | 60 | $155.76 \pm 3.91$ |
| 14 | 0.3 | 100 | 1 | 72 | $153.01 \pm 2.99$ |
| 15 | 0.6 | 50 | 1 | 48 | $116.705 \pm 2.98$ |
| 16 | 0.45 | 75 | 1.5 | 60 | $127.023 \pm 3.01$ |
| 17 | 0.45 | 75 | 2.5 | 60 | $161.495 \pm 4.23$ |
| 18 | 0.45 | 125 | 1.5 | 60 | $102.909 \pm 2.74$ |
| 19 | 0.6 | 100 | 1 | 72 | $226.426 \pm 5.67$ |
| 20 | 0.45 | 75 | 1.5 | 84 | $232.456 \pm 5.98$ |
| 21 | 0.45 | 75 | 1.5 | 60 | $176.324 \pm 3.79$ |
| 22 | 0.45 | 75 | 0.5 | 60 | $154.256 \pm 2.77$ |
| 23 | 0.3 | 50 | 1 | 72 | $203.134 \pm 4.58$ |
| 24 | 0.6 | 100 | 2 | 72 | $190.135 \pm 3.54$ |
| 25 | 0.75 | 75 | 1.5 | 60 | $167.646 \pm 3.47$ |
| 26 | 0.3 | 100 | 2 | 48 | $101.44 \pm 2.32$ |
| 27 | 0.6 | 50 | 1 | 72 | $205.286 \pm 4.02$ |
| 28 | 0.45 | 75 | 1.5 | 60 | $171.822 \pm 3.26$ |
| 29 | 0.45 | 75 | 1.5 | 60 | $147.745 \pm 3.26$ |
| 30 | 0.3 | 50 | 2 | 72 | $214.525 \pm 3.79$ |

Statistical analysis of the linear model design showed a model F-value that equaled 12.98, which implies that the model is significant. The values of "Prob. > F" lower than 0.05 indicate that the model terms are significant. Peptone, aeration, CaCl$_2$, and the incubation period were significant model terms. The analysis revealed that $R^2_{Pred}$ equaled 0.9238, and was comparable to the $R^2_{Adj}$ (0.8526). The precision ratio (Adeq) of 14.725 indicates

an adequate signal. We analyzed CCD results via ANOVA, which yielded the following regression equation for the level of alpha-amylase production.

$$
\begin{aligned}
\alpha - \text{amylase activity} \\
= +2326.27 + 41.28 \times \text{A} - 42.94 \times \text{B} + 50.80 \times \text{C} + 686.32 \times \text{D} \\
= +128.37 \times \text{A} \times \text{B} \, 163.15 \times \text{A} \times \text{C} + 31.95 \times \text{A} \times \text{D} + 29.78 \times \text{B} \times \text{C} \, 12.53 \times \text{B} \times \text{D} \\
= +32.66 \times \text{C} \times \text{D} + 73.77 \times \text{A2} \, 178.33 \times \text{B2} + 17.99 \times \text{C2} + 115.33 \times \text{D2}
\end{aligned}
\tag{1}
$$

where A = starch, B = peptone, C = yeast extract, and D = aeration.

The production of alpha-amylase enzyme was elevated using the following combinations of variables: aeration–peptone; $CaCl_2$–peptone; incubation period–peptone; $CaCl_2$–aeration, and incubation period–aeration.

### 3.5. Gamma Irradiation Enhanced the Production of Alpha-Amylase

The freshly cultured MS009 slants were irradiated with different doses of gamma irradiation. The extracellular production of $\alpha$-amylase by MS009 increased when increasing the dose of gamma irradiation from 1 to 10 kGy. A significant increase in the specific alpha-amylase activity ($268.25 \pm 1.03$ U/mg) was detected at a gamma irradiation dose of 6 KGy $p \leq 0.05$ (Figure 3). Gamma irradiation of MS009 at 6 KGy resulted in a significant increase in alpha-amylase activity ($319.45 \pm 2.05$ U/mg $p \leq 0.05$) when grown under the optimal reaction conditions. Partial purification of $\alpha$-amylase, from the crude enzyme extracts resulting from submerged fermentation, was performed using $(NH_4)_2SO_4$ and acetone precipitation, which resulted in 1.5-fold purification when using acetone precipitation (Table 6).

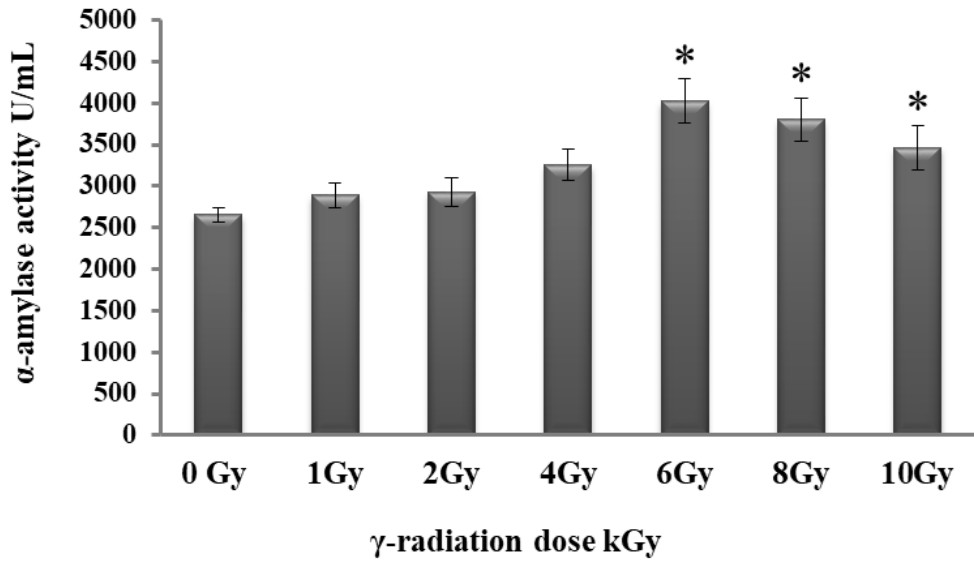

**Figure 3.** Effects of gamma irradiation on the production of alpha-amylase enzyme by MS009; * $p \leq 0.05$.

**Table 6.** Alpha-amylase-specific activity using different purification methods.

| Purification Method | Amylase-Specific Activity (U/mg) | Purification Factor |
|---|---|---|
| Crude enzyme | $68.59 \pm 2.97$ | 1 |
| $(NH_4)_2SO_4$ (40%) | $70.985 \pm 3.68$ | 1.03 |
| $(NH_4)_2SO_4$ (50%) | $74.42 \pm 4.46$ | 1.08 |
| $(NH_4)_2SO_4$ (60%) | $82.5 \pm 4.14$ | 1.2 |
| $(NH_4)_2SO_4$ (70%) | $94.75 \pm 5.14$ | 1.4 |
| $(NH_4)_2SO_4$ (80%) | $75.1 \pm 4.38$ | 1.1 |
| Acetone | $100.88 \pm 5.64$ | 1.5 |

Data are expressed as mean values $\pm$ S.E.M. All of the results showed significant differences between treatments (Duncan's test, $p = 0.05$).

### 3.6. XRD, FTIR, and SEM/EDAX Characterization of Alpha-Amylase-CLBFNPs

The prepared $\alpha$-amylase-CLBFNPs were characterized by different techniques. XRD analysis determined the crystal composition and the average crystal size of the synthesized alpha-amylase-CLBFNPs. The crystal and/or amorphous compositions of the starting primary materials (chitosan, barium ferrite, and alpha-amylase) and of the synthesized alpha-amylase-CLBFNPs are shown in Figure 4. Chitosan presented $2\theta$ at $11.45°$ and $20.52°$, which were similar to the amorphous type of chitosan; alpha-amylase presented $2\theta$ at $21.78°$. The XRD data for barium ferrite were investigated at $2\theta$ of $32.45°$, $35.14°$, $45.87°$, $53.84°$, $57.85°$, and $63.52°$, accounting for the Bragg reflections at (110), (114), (206), (209), (217), and (220), respectively. The results of the XRD data of the synthesized alpha-amylase-BFCMNPs show the diffraction characteristics, including $2\theta$ at $31.85°$, $35.16°$, $46.74°$, $57.89°$, and $62.54°$, which describe the Bragg reflections at (110), (114), (206), (217), and (220), respectively.

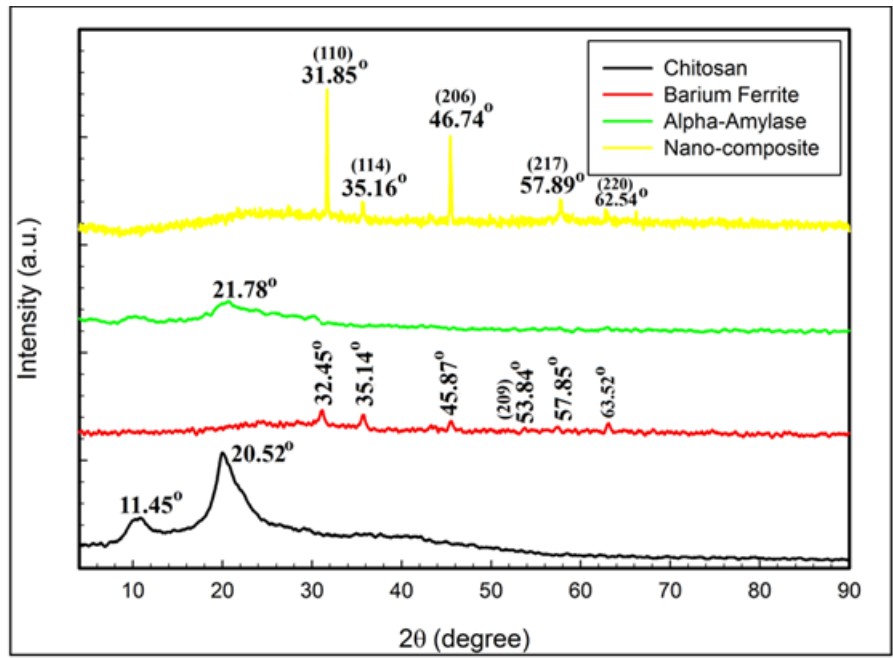

**Figure 4.** XRD patterns of chitosan at $11.45°$ and $20.52°$; barium ferrite at $32.45°$, $35.14°$, $45.87°$, $53.84°$, $57.85°$, and $63.52°$, accounting for the Bragg reflections at (110), (114), (206), (209), (217), and (220), respectively; alpha-amylase at $21.78°$; and synthesized alpha-amylase-BFCMNPs at $31.85°$, $35.16°$, $46.74°$, $57.89°$, and $62.54°$, which describe the Bragg reflections at (110), (114), (206), (217), and (220), respectively.

The common crystallite size of the incorporated synthesized alpha-amylase-CLBFNPs was determined using the Williamson–Hall (W–H) equation, and equaled 38.25 nm for the nanocomposite, according to the following equation:

$$\beta \cos \theta = \frac{k\lambda}{D_{W-H}} + 4\varepsilon \sin \theta$$

where DW − H is the average crystallite size, $\beta$ is the full width at half-maximum, $\lambda$ is the X-ray wavelength, $\theta$ is the Bragg angle, $k$ is a constant, and $\varepsilon$ is the strain of the samples.

The uniform features and the surface morphology of the synthesized alpha-amylase-CLBFNPs were determined by HRSEM imaging (Figure 5); barium ferrite NPs were separated as bright spherical particles in the core, while the alpha-amylase with chitosan appeared as a condensed blackish material.

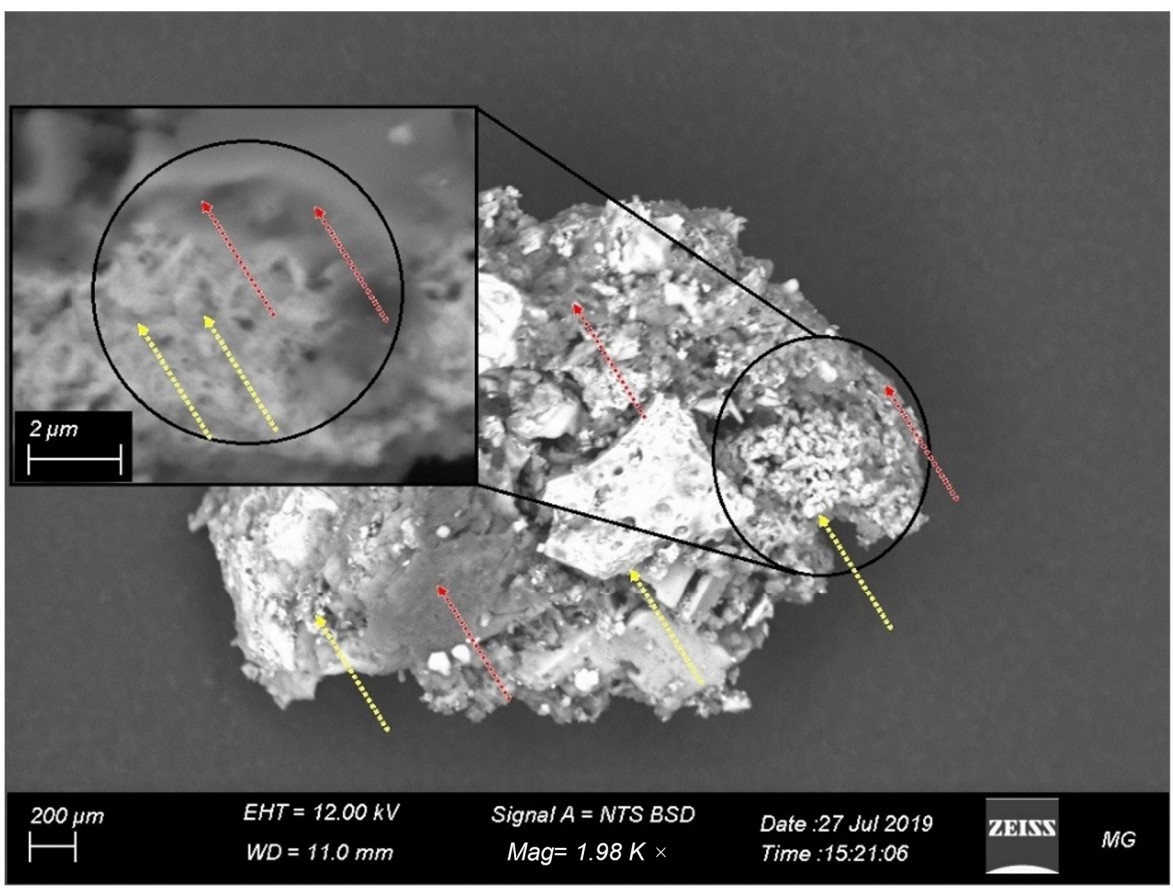

**Figure 5.** FE-SEM image of the synthesized alpha-amylase doped on chitosan-loaded barium ferrite nanoparticles. The arrows indicate the porous structure of chitosan-loaded barium ferrite, where alpha-amylase is deposited.

EDX examinations verified that the synthesized alpha-amylase-CLBFNPs were stoichiometric, with a common typical form. The typical X-ray peaks of C, N, O, Na, Si, P, Cl, P, Ca, Fe, and Ba atoms were obvious in the EDX of synthesized alpha-amylase-BFCMNPs. The C, N, O, Na, Si, P, Cl, P, and Ca atoms corresponded to glutaraldehyde, alpha-amylase, chitosan, and sodium phosphate buffer components. Elemental mapping images of the synthesized alpha-amylase-BFCMNPs, showing C, N, O, Na, Si, P, Cl, P, Ca, Fe, and Ba components, are shown in Figure 6. The presence of Ba and Fe atoms related to barium ferrite appeared as condensed bright distribution (pink and turquoise colors, respectively) (Figure 7). The C, N, O, Na, Si, P, Cl, P, and Ca atoms were for glutaraldehyde

used in crosslinking during the immobilization, alpha-amylase, chitosan, and sodium phosphate buffer.

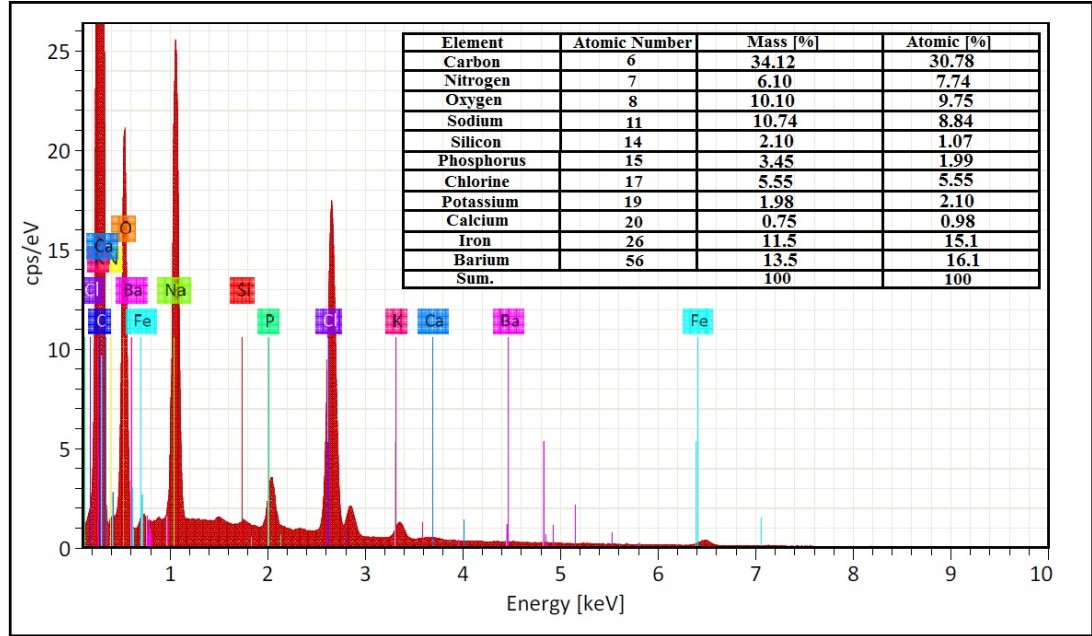

**Figure 6.** Elemental analysis of the synthesized alpha-amylase doped on chitosan-loaded barium ferrite nanoparticles. The typical X-ray peaks of C, N, O, Na, Si, P, Cl, P, Ca, Fe, and Ba atoms are observable in the EDX of synthesized alpha-amylase-BFCMNPs. The C, N, O, Na, Si, P, Cl, P, and Ca atoms correspond to glutaraldehyde, alpha-amylase, chitosan, and sodium phosphate buffer components.

FTIR investigations were conducted to define the interactions within the synthesized alpha-amylase-CLBFNPs (Figure 8). The FTIR spectrum of alpha-amylase had absorption bands at 3257.68, 2980.48, 2867.44, 1644.88, 1420.24, 1296.4, 1067.44, and 985.36 cm$^{-1}$; the absorption bands for barium ferrite were 3332.56, 2980.48, 2867.44, 1667.2, 1423.84, 1303.6, 1075.36, and 921.28 cm$^{-1}$; the absorption bands for chitosan were at 3264.88, 2980.48, 2867.44, 1644.88, 1423.84, 1296.4, 1063.84, and 943.6 cm$^{-1}$; and the absorption bands for the synthesized alpha-amylase-CLBFNPs were 3317.44, 2980.48, 2867.44, 1637.68, 1442.56, 1292.8, 1067.44, and 928.48 cm$^{-1}$. The extended peaks at 3257.68, 3332.56, 3264.88, and 3317.44 were attributed to –OH of the hydroxyl group, and the peaks at 2980.48 and 2867.44 cm$^{-1}$ were assigned to the symmetric and asymmetric –CH vibration of the –CH$_2$ group. The observed peaks at 1644.88, 1667.2, and 1637.68 cm$^{-1}$ were because of –C=O stretching of the ester group. The extended peaks at 1420.24, 1423.84, and 1442.56 were attributed to -NH stretching. The peaks at 1420.24, 1423.84, and 1442.55cm$^{-1}$ were designated to –NH. The extended peaks at 1296.40, 1303.6, and 1292.8 cm$^{-1}$ were attributed to –CH stretching. Additional bands at 1067.44, 1075.36, and 1063.84 cm$^{-1}$ were designated to C-O. Moreover, the peaks at 985.36, 921.28, 943.6, and 928.48 cm$^{-1}$ were due to C-O. Our FTIR conclusions are comparable with the literature.

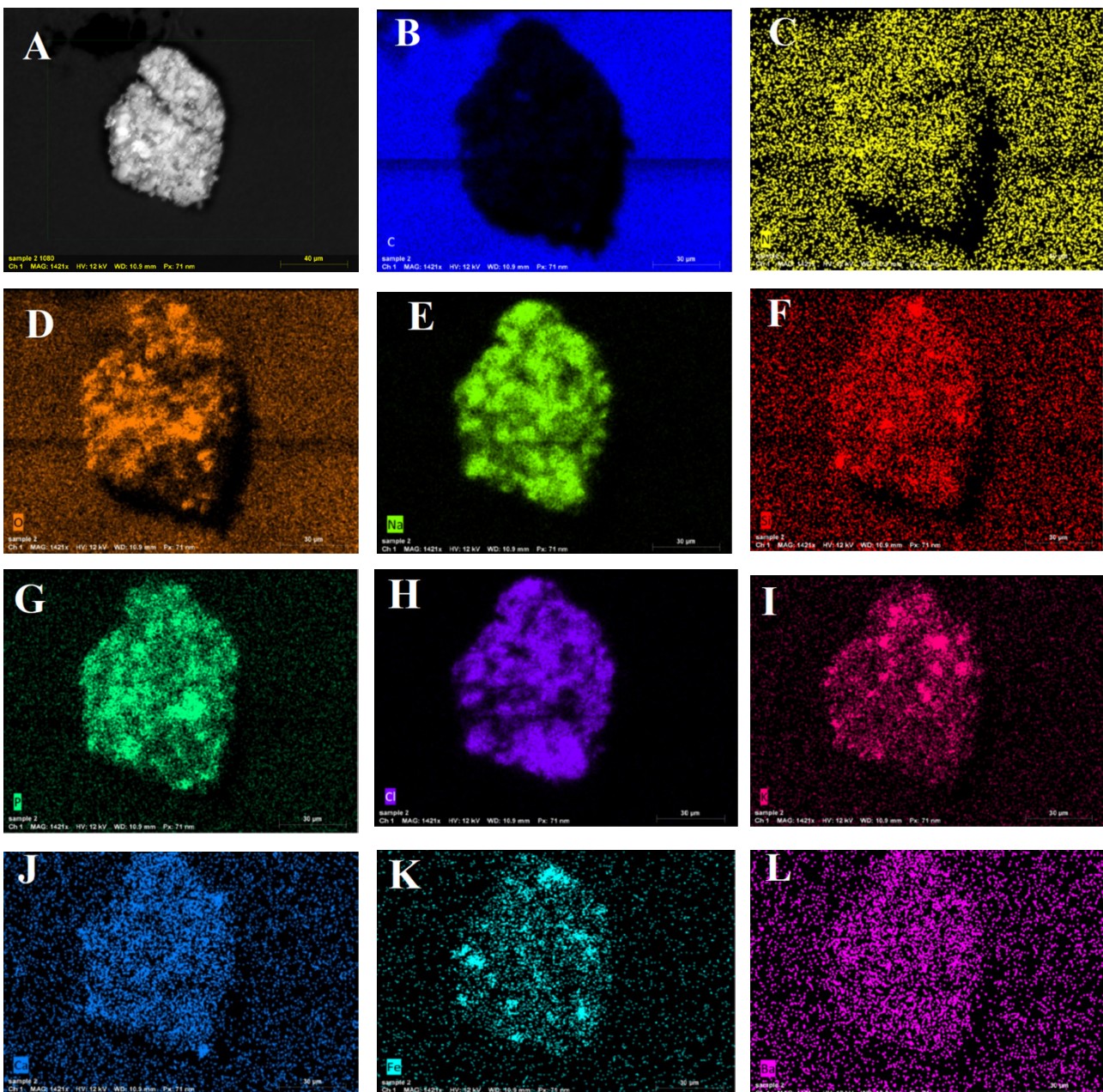

**Figure 7.** SEM/EDX mapping images of the synthesized alpha-amylase doped on chitosan-loaded barium ferrite nanoparticles. (**A**) Alpha-amylase-CLBFNPs, (**B**) Carbon, (**C**) nitrogen, (**D**) oxygen, (**E**) sodium, (**F**) silicon, (**G**) phosphorus, (**H**) chlorine, (**I**) potassium, and (**J**) calcium atoms were for glutaraldehyde used in crosslinking during the immobilization, alpha-amylase, chitosan, and sodium phosphate buffer. The presence of (**L**) Ba and (**K**) Fe atoms of barium ferrite appears as condensed bright distribution (pink and turquoise colors, respectively).

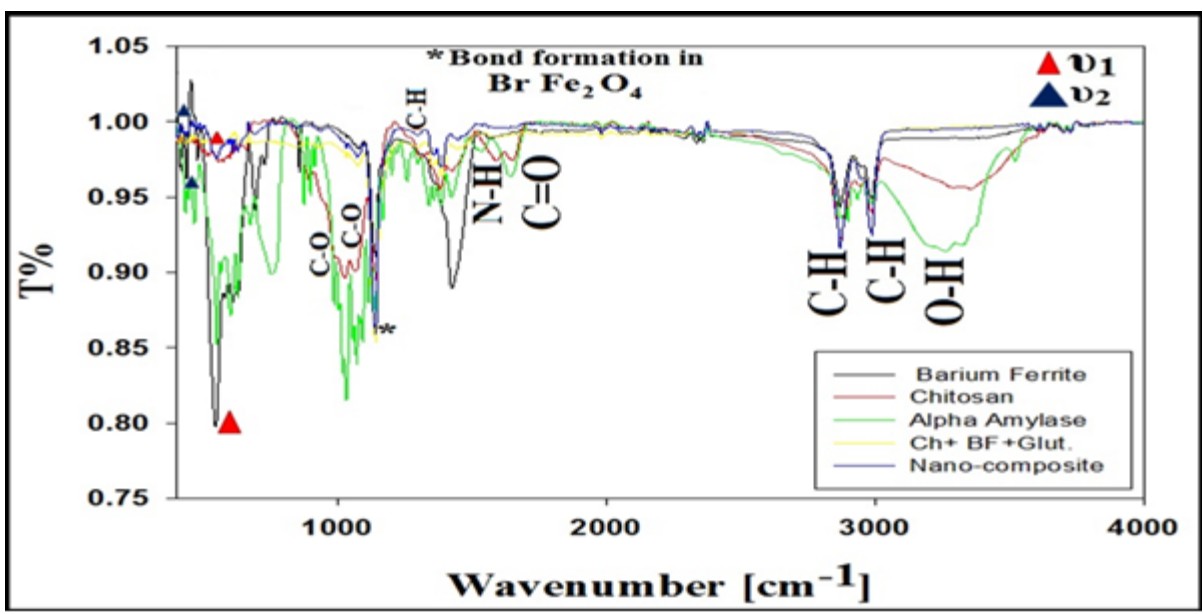

**Figure 8.** FTIR spectra of barium ferrite, chitosan, alpha-amylase, and chitosan–barium ferrite, nanocomposite (chitosan-loaded barium ferrite nanoparticles). The alpha-amylase absorption bands are at 3257.68, 2980.48, 2867.44, 1644.88, 1420.24, 1296.4, 1067.44, and 985.36 cm$^{-1}$; the absorption bands for barium ferrite are at 3332.56, 2980.48, 2867.44, 1667.2, 1423.84, 1303.6, 1075.36, and 921.28 cm$^{-1}$; the absorption bands for chitosan are at 3264.88, 2980.48, 2867.44, 1644.88, 1423.84, 1296.4, 1063.84, and 943.6 cm$^{-1}$; and the absorption bands for the synthesized alpha-amylase-CLBFNPs are at 3317.44, 2980.48, 2867.44, 1637.68, 1442.56, 1292.8, 1067.44, and 928.48 cm$^{-1}$. * Represent bond formation between barium ferrite, chitosan and alpha amylase.

### 3.7. The Synthesized Alpha-Amylase-CLBFNPs Are Active and Reusable

A significant increase in the specific activity of the prepared α-amylase-CLBFNPs (246.85 U/mg) compared to free alpha-amylase (177.12 U/mg) was detected ($p \leq 0.05$). The alpha-amylase-CLBFNP beads retained activity for up to five cycles of usage, showing a subsequent decrease in activity after every cycle. An alpha-amylase-specific activity of 90% was detected after the fifth cycle of usage of α-amylase-CLBFNPs. We measured the activity of both the free and immobilized α-amylase at different working temperatures and pH values. The immobilized α-amylase showed higher activity than the free α-amylase at all tested pH values, with the highest activity recorded at pH 7–8, according to Duncan's multiple range test (Figure 9). The immobilized enzyme showed higher activity than the free enzyme at all tested temperatures, with the highest activity recorded at temperatures ranging from 50 to 60 °C, according to Duncan's multiple range test (Figure 10).

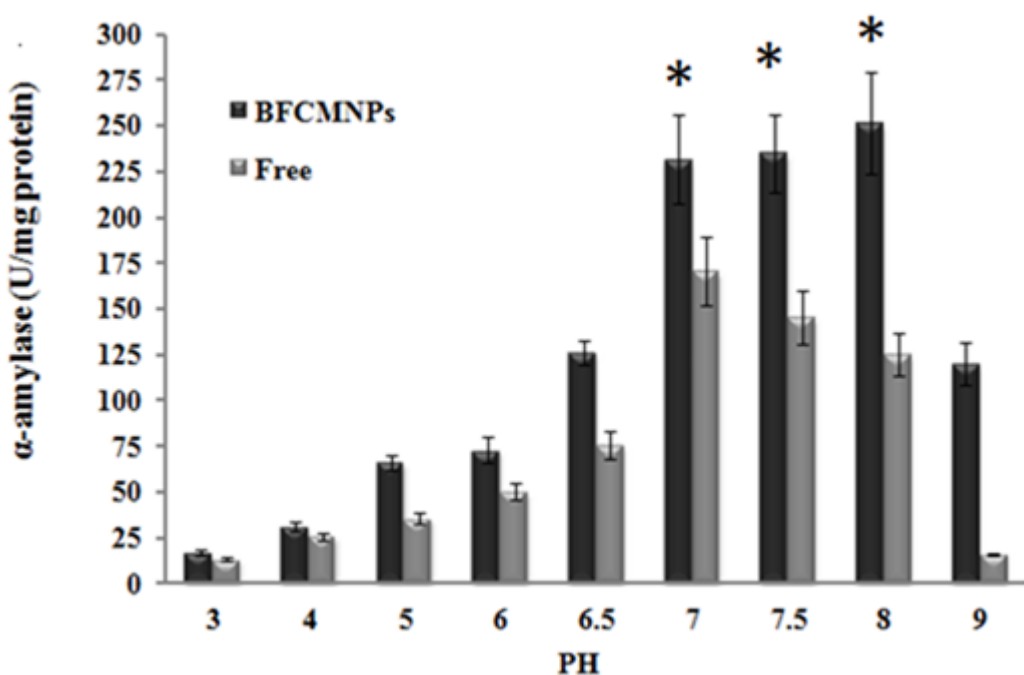

**Figure 9.** Effect of pH on the specific activity of free and immobilized alpha-amylase-BFCMNPs; * $p \leq 0.05$.

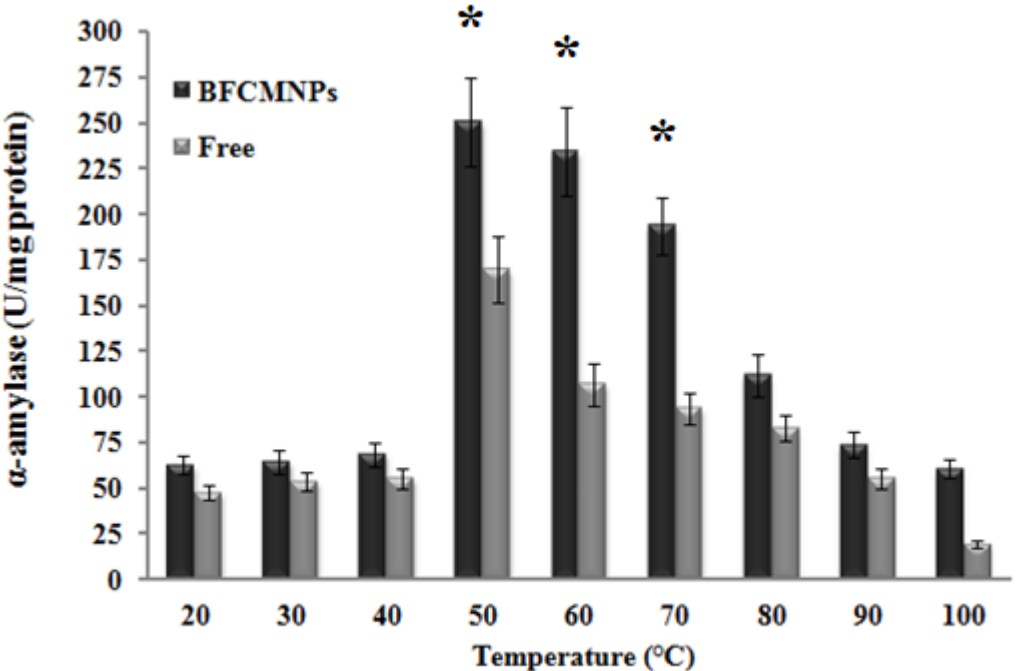

**Figure 10.** Effect of temperature on the specific activity of free and immobilized alpha-amylase-BFCMNPs; * $p \leq 0.05$.

## 4. Discussion

Microbial amylases have many industrial applications, as they are of higher stability, with great genetic diversity, high enzymatic activity in a wide range of extreme conditions, simple and cost-effective production, and easy manipulation to obtain enzymes with the desired characteristics [53]. MNPs are efficient enzyme carriers because of their biocompatibility, stability, high surface-area-to-volume ratio, and superparamagnetic properties [30].

We isolated and identified an alpha-amylase producer from soil, optimized its production using P–BD, CCD, and gamma irradiation, and prepared alpha-amylase-CLBFNP beads.

Screening for alpha-amylase production was carried out by sub-culturing on starch agar. This is a rapid screening method for the detection of alpha-amylase production [4,39]. We collected soil samples from agricultural fields of different crops. As reported in the literature, the level of amylase production by a given microorganism depends on the microbe's origin, where isolates from starch- or amylose-rich environments naturally produce higher amounts of amylase enzymes [54,55]. This is consistent with our findings, as MS009 (the potent producer) was isolated from the agricultural soil of a guava field. Guava seeds are rich in carbohydrates, lipids, and proteins; the soil in a guava field contains 23.6 g of carbohydrates per cup [56]. In accordance with previous studies, we used the DNS assay to further confirm alpha-amylase production [57].

In our study, alpha-amylase producers were either bacterial or fungal isolates. MS009 was identified as *Bacillus paramycoides* using 16S rRNA sequencing and phylogenetic mapping. *Bacillus* species produce a large variety of extracellular enzymes, including amylases, which have industrial importance [58]. The level of amylase production varies between microorganisms from the same genus, species, and strain, but its production by bacteria is cheaper and faster than that by other microorganisms [56]. Several bacterial species can produce alpha-amylase for industrial applications. Most alpha-amylase-producing bacteria belong to *Bacillus* species, including *B. subtilis*, *B. amyloliquefaciens*, *B. licheniformis*, *B. stearothermophilus*, *B. coagulans*, *B. polymyxa*, *B. mesentericus*, *B. vulgaris*, *B. megaterium*, *B. cereus*, *B. dipsosauri*, and *B. halodurans* [56,59]. Other microbial sources for alpha-amylase production for commercial purposes include fungi such as *Aspergillus* spp., *Penicillium* spp., *Streptomyces rimosus*, *Thermomyces lanuginosus*, *Pycnoporus sanguineus*, *Cryptococcus flavus*, *Thermomonospora curvata*, and *Mucor* spp. [55].

Plackett–Burman design, followed by CCD, was used to optimize the critical factors required for maximum alpha-amylase production by MS009. P–BD was previously used to select the factors that affect the production of $\alpha$-amylase by *Aspergillus niger* ATCC16404 [44]. An $\alpha$-amylase production of $222.254 \pm 5.07$ U/mg was achieved using the P–BD. The following parameters had a significant effect on alpha-amylase production, including the use of peptone as a nitrogen source, aeration (volume of culture media/volume of flask), supplementing the fermentation medium with $CaCl_2$, the incubation period, and inoculum size. CCD maximizes the amount of information that can be obtained, while limiting the numbers of individual experiments [44]. We achieved a maximum $\alpha$-amylase production of $232.456 \pm 5.98$ U/mg using the CCD design. The reactions' surfaces analysis shows that the following combinations of factors—(aeration and peptone), ($CaCl_2$ and peptone), (incubation period and peptone), ($CaCl_2$ and aeration), and (incubation period and aeration)—are the most influential to increase the reaction. This confirms the previous data obtained from analysis of P–BD results, and strongly suggests that these variables play a significant role in $\alpha$-amylase production. The decrease in enzyme production by changing the conditions—other than the optimum—in the optimization process may be due to the denaturation or decomposition of $\alpha$-amylase as a result of the interaction with other components in the culture medium [60,61].

We recorded a significant increase in alpha-amylase production following gamma irradiation of MS009 with 6 kGy ($319.45 \pm 2.05$ U/mg, $p \leq 0.05$), when cultured using the optimized culturing conditions. Low-dose irradiation may induce mutations in microorganisms that could improve the productivity of antibiotics, organic acids, amino acids, vitamins, alcohols, pigments, and enzymes [33,61]. Exposing *Bacillus subtilis* to different doses of gamma radiation, ranging from 1 to 6 kGy, resulted in a slightly enhanced $\alpha$-amylase activity compared to non-irradiated cells [62]. Gamma irradiation was previously reported to increase extracellular lipase production by *A. niger* at a dose of 1.4 kGy [33,63–65].

In our study, alpha-amylase was immobilized on chitosan-coated barium ferrite nanoparticles. Enzyme immobilization on magnetic nanoparticles has been reported in previous studies, e.g., alpha-amylase on gum-acacia-stabilized magnetite nanoparticles [66];

pullulanase enzymes on hybrid $Fe_3O_4$–chitosan nanoparticles [67]; lipase enzymes on magnetic nanoparticles [33], and catalase on chitosan/ZnO/$Fe_3O_4$ nanoparticles [68]. We reported a 139% increase in the specific activity of the prepared $\alpha$-amylase-CLBFNPs compared to the free amylase. This is in contrast to previous studies that reported retaining of 88.37% specific activity for alpha-amylase immobilized on magnetic nanoparticles compared to the free enzyme [69]. In our study, the binding between the hydrophobic particles of barium ferrite and the hydrophobic area of amylase took place via interfacial activation, and several techniques were used to characterize enzyme immobilization. XRD, SEM, EDX, and FTIR were previously reported to characterize enzyme immobilization on nanoparticles [48,50,51]. FTIR verified the production of the capped barium ferrite NPs; broad bands around 3400 cm$^{-1}$ corresponded to stretching vibrations of –OH from the chitosan coating on the MNPs [70]. The strong absorption peak at 1626 cm$^{-1}$, corresponding to the –NH$_2$ groups from chitosan, confirmed the surface coating of the MNPs by chitosan [71–73]. The absorption peak at 1640 cm$^{-1}$ in the $\alpha$-amylase spectra, related to the (CO-NH) amide band, also appeared in the amylase-CLBFNPs' spectra [74,75]. This confirms the successful immobilization of $\alpha$-amylase on chitosan-coated MNPs.

An enzyme can change its structural conformation, which may diminish or enhance its stability and physicochemical properties [69,76,77]. The effective catalytic activity of the enzyme at variable pH and temperature values is the main purpose of enzyme immobilization. The stability and activity of the prepared $\alpha$-amylase-CLBFNPs composite were tested under different physicochemical conditions. Immobilization resulted in a slowdown in the denaturing effect of increasing pH and/or temperature, which was exhibited by a one-unit shift to the alkaline side and a +10 °C shift in temperature, showing enhanced stability. This is in accordance with previous studies that suggested that immobilization makes the active sites of lipase more exposed to solvents than the folded/dissolved form. In our study, 90% activity was retained for up to five cycles of use of alpha-amylase-CLBFNPs. This was superior to previous studies, which reported lower activity for amylase magnetic nanoparticles (87% [51] and 60% [48]) after five cycles of usage; gum-acacia-stabilized magnetite nanoparticles retained 70% activity for up to six cycles of usage [66]. The decrease in activity after every cycle of usage may be attributed to the denaturation of the enzyme, or to the physical loss of the enzyme from its carrier [49].

## 5. Conclusions

MS009, a *Bacillus paramycoides* isolate, is a potent alpha-amylase producer recovered from Egyptian agricultural soil. Optimization of culture conditions using P–BD, CCD, and gamma irradiation resulted in a significant increase in alpha-amylase production. The immobilization of $\alpha$-amylase on CLBFNPs gave our product additional characteristics, such as selective and easy enzyme recovery from the medium under magnetic force, which helps in repetitive and continuous use, localization of the interaction, prevention of product contamination, and reducing effluent problems and material handling.

**Author Contributions:** Conceptualization, O.H., A.-E.S., A.E.-B. and M.R.; methodology, M.H.; software, M.H., O.H. and A.E.-B.; validation, M.H., O.H. and A.E.-B.; formal analysis, M.H., O.H. and A.E.-B.; investigation, M.H., O.H. and A.E.-B.; resources, M.H., O.H. and A.E.-B.; data curation, M.H., O.H. and A.E.-B.; writing—original draft preparation, M.H.; writing—review and editing, O.H., A.E.-B. and M.R.; visualization, M.H., O.H. and A.E.-B.; supervision, O.H., A.-E.S., A.E.-B. and M.R.; project administration, A.E.-B. All authors have read and agreed to the published version of the manuscript.

**Funding:** This research received no external funding.

**Data Availability Statement:** Not applicable.

**Acknowledgments:** The authors would like to thank the Nanotechnology Research Unit, Drug Microbiology Lab., Drug Radiation Research Department, NCRRT, Egypt. The authors would also like to thank Mohamed Gobara (Military Technical College, Egyptian Armed Forces) and the Zeiss microscope team in Cairo for their invaluable advice during this study.

**Conflicts of Interest:** The authors declare no conflict of interest.

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
