# Peer review of "Optimization of Alpha-Amylase Production by a Local Bacillus paramycoides Isolate and Immobilization on Chitosan-Loaded Barium Ferrite Nanoparticles"

_fermentation, doi:10.3390/fermentation8050241_

Round 1
Reviewer 1 Report
The manuscript: “optimization of alpha amylase production from Bacillus paramycoides and its immobilization on chitosan loaded barium ferrite nanoparticles”, describes the isolation of microorganisms from Egyptian soil samples and the determination of their capacity to produce alpha amylases. Based in the previous study, the strain MS009, able to produce the maximum alpha amylase activity (under specific parameters) was selected to realize an optimization study of the enzyme production. MS009 was submitted of different steps of mutation using gamma irradiation, in order to enhance the alpha amylase production. After that, the alpha amylase from MS009 was partially purified and immobilized on chitosan-loaded barium ferrite 185 nanoparticles (CLBFNPS). The CLBFNPS characterization by X-ray diffraction, fourier transform infrared and scanning electron microscopy were widely described. Finally, the authors demonstrated that the synthesized alpha amylase-CLBFNPS are active and reusable.
I consider that this report has several highlights, however, due to the authors trying to cover different issues in this work, the paper loses clarity and is difficult to understand.
First of all, the authors must redesign the introduction section because in its present form looks disorganized. These paragraphs must follow a continuity of ideas for the reader to clearly understand what the objective and the contributions of the work are.
In the same way, the results are written without a previous explanation of the purpose of each experiment, leaving the reader confused. If the authors describe why the made the experiment more clearly the reader could understand the results and their contributions. I consider that the section of results and discussion must be put together in order to be clearer.
Specific comments
-Table 1, say “mgm”, instead “mg”
-Line 266 the authors must mention the variables that they tested in the results section (pH, Temp, agitation, etc.)
-Figure 3 has low quality, it’s impossible to see.
-It is necessary to determine de purity percentage of the enzyme partially purified. A figure showing the enzyme is important.
I consider that this manuscript requires extensive modifications in multiple sections. I do not recommend its publication in the journal “Fermentation” in its current state.
Author Response
We would like to thank the reviewer for taking the time and effort necessary to review the manuscript. We sincerely appreciate all valuable comments and suggestions, which helped us to improve the quality of the manuscript.
Comments and Suggestions for Authors
The manuscript: “optimization of alpha amylase production from Bacillus paramycoides and its immobilization on chitosan loaded barium ferrite nanoparticles”, describes the isolation of microorganisms from Egyptian soil samples and the determination of their capacity to produce alpha amylases. Based in the previous study, the strain MS009, able to produce the maximum alpha amylase activity (under specific parameters) was selected to realize an optimization study of the enzyme production. MS009 was submitted of different steps of mutation using gamma irradiation, in order to enhance the alpha amylase production. After that, the alpha amylase from MS009 was partially purified and immobilized on chitosan-loaded barium ferrite nanoparticles (CLBFNPS). The CLBFNPS characterization by X-ray diffraction, fourier transform infrared and scanning electron microscopy were widely described. Finally, the authors demonstrated that the synthesized alpha amylase-CLBFNPS are active and reusable.
I consider that this report has several highlights, however, due to the authors trying to cover different issues in this work, the paper loses clarity and is difficult to understand.
1-First of all, the authors must redesign the introduction section because in its present form looks disorganized. These paragraphs must follow a continuity of ideas for the reader to clearly understand what the objective and the contributions of the work are.
We would like to thank the reviewer for his comment. The introduction was rewritten to clarify the idea.
2- In the same way, the results are written without a previous explanation of the purpose of each experiment, leaving the reader confused. If the authors describe why the made the experiment more clearly the reader could understand the results and their contributions. I consider that the section of results and discussion must be put together in order to be clearer.
We would like to thank the reviewer for his comment. A brief for the reason for doing each experiment was added to help in understanding the results.
Specific comments
3-Table 1, say “mgm”, instead “mg”
We would like to thank the reviewer for his comment and it is corrected in table 1.
4-Line 266 the authors must mention the variables that they tested in the results section (pH, Temp, agitation, etc.)
We would like to thank the reviewer for his comment. The tested variables are now presented in a new table named table 1 and table 2 in the methods section and tables 4 and 5 in the results section.
5-Figure 3 has low quality, it’s impossible to see.
We would like to thank the reviewer for his comment. Figure 3 is deleted.
6-It is necessary to determine de purity percentage of the enzyme partially purified. A figure showing the enzyme is important.
We would like to thank the reviewer for his comment. This was added as a separate column in table 6.
Reviewer 2 Report
The paper is interesting and it explores the isolation of new species for industrial enzyme production. However, in order to make it a stronger paper, some improvement is necessary. All Figures need to be improved, their quality is low and difficult to read. I suggest some other points before it can be published:
- A strong English revision by a professional or native speaker is necessary;
- Introduction, lines 49 to 80: why to present such information into numbered topics? A well constructed introduction would make the manuscript more fluid to read.
- If amylases already are about to 25% of enzyme market, why to develop the present work? The introduction needs to be improved to make clear the novelty of the work. What is the importance to isolate new microrganisms that produce amylase? How is the state of the art of amylases immobilization into magnetic particles?
- Scientific names must be in italic font. Please check the entire manuscript (Item 2.1 and line 255, for example)
- Item 2.4 please include the tables S1 and S2 in the main manuscript text and not as supplementary file. It is also important to provide tables with the DOES codded variables.
- Item 2.7: is it 25 mg or 25g? Please check if the information in the text is correct
- Item 2.9 please provide the buffer used for each pH value in the assays
- Please improve Figures 2 and 3 quality, it is not possible to read as it is.
- The use of incubation period is not usual in DOE, once it is already expected to improve enzyme production over time. How authors justify the choice of this variable?
- Line 296-302: the paragraph is not justified as requested by journals guidelines
- What information is given by figure 3? The model has a significant curvature? The figure only shows flat charts. It could be removed from the manuscript.
- Table 2 what is the purification factor of each condition?
- Lines 346-348: please set this in methodology section
- Lines 380-384: is it the legend of the figure 8? If so, please format it according to journals guideline
- Item 3.7: authors say “A significant increase in the activity of the immobilized alpha-amylase (246.85 U/mg) 413 compared to free alpha-amylase (177.12 U/mg)”. Are both activities measured in U/mg of protein? If yes, what is the CLBFNPS activity in U/mg of support? This is an important information to compare to other alfa amylases derivatives in literature and should be discussed in discussion section.
- The results about the immobilization of the enzyme into the magnetic particles are missing, what was the immobilization yield and recovered activity? Please add the results of these experiments.
Author Response
We would like to thank the reviewer for taking the time and effort necessary to review the manuscript. We sincerely appreciate all valuable comments and suggestions, which helped us to improve the quality of the manuscript.
Comments and Suggestions for Authors
The paper is interesting and it explores the isolation of new species for industrial enzyme production. However, in order to make it a stronger paper, some improvement is necessary. All Figures need to be improved, their quality is low and difficult to read. I suggest some other points before it can be published:
- A strong English revision by a professional or native speaker is necessary;
We would like to thank the reviewer for his comment. English language was revised by a professional colleague.
- Introduction, lines 49 to 80: why to present such information into numbered topics? A well constructed introduction would make the manuscript more fluid to read.
We would like to thank the reviewer for his comments. This was a formatting problem and is corrected now.
- If amylases already are about to 25% of enzyme market, why to develop the present work? The introduction needs to be improved to make clear the novelty of the work. What is the importance to isolate new microrganisms that produce amylase? How is the state of the art of amylases immobilization into magnetic particles?
We would like to thank the reviewer for his comment. The introduction was rewritten to clarify the idea much better.
- Scientific names must be in italic font. Please check the entire manuscript (Item 2.1 and line 255, for example)
We would like to thank the reviewer for his comment. All scientific names were capitalized all throughout the manuscript.
- Item 2.4 please include the tables S1 and S2 in the main manuscript text and not as supplementary file. It is also important to provide tables with the DOES codded variables.
We would like to thank the reviewer for his comment. The Supplementary tables are incorporated in the main text as tables 4 and 5 respectively. Tables for the used variables were presented in text as tables 1 and 2.
- Item 2.7: is it 25 mg or 25g? Please check if the information in the text is correct
We would like to thank the reviewer for his comment. The information in the text is correct.
- Item 2.9 please provide the buffer used for each pH value in the assays
We would like to thank the reviewer for his comment. We used PBS in all reactions and the pH was adjusted with 0.1N HCl and 0.1N NaOH to the desired pH and this was added in the methods section (line 199).
- Please improve Figures 2 and 3 quality, it is not possible to read as it is.
We would like to thank the reviewer for his comment. Figure 3 is deleted and the quality of Figure 2 is improved.
- The use of incubation period is not usual in DOE, once it is already expected to improve enzyme production over time. How authors justify the choice of this variable?
We would like to thank the reviewer for his comment. Incubation period is one of the studied variables in previous studies. Prolonging the incubation period result in an increase in production but beyond a certain limit the increase in production won’t be significant and hence increase the cost of the process. So choosing the optimum incubation period is an important factor.
- Line 296-302: the paragraph is not justified as requested by journals guidelines
We would like to thank the reviewer for his comment. We adjusted it.
- What information is given by figure 3? The model has a significant curvature? The figure only shows flat charts. It could be removed from the manuscript.
We would like to thank the reviewer for his comment. Figure 3 was deleted.
- Table 2 what is the purification factor of each condition?
We would like to thank the reviewer for his comment. The purification fold was calculated and added as an extra column in the table, (Now table 6).
- Lines 346-348: please set this in methodology section
We would like to thank the reviewer for his comment. This is the figure legend and was correctly formatted.
- Lines 380-384: is it the legend of the figure 8? If so, please format it according to journals guideline
We would like to thank the reviewer for his comment. This is corrected according to the journal guidelines.
- Item 3.7: authors say “A significant increase in the activity of the immobilized alpha-amylase (246.85 U/mg) 413 compared to free alpha-amylase (177.12 U/mg)”. Are both activities measured in U/mg of protein? If yes, what is the CLBFNPS activity in U/mg of support? This is an important information to compare to other alfa amylases derivatives in literature and should be discussed in discussion section.
We would like to thank the reviewer for his comment. Yes the activity is measured in U/mg and the activity of enzyme immobilized on CLBFNPS is (246.85 U/mg) and this was discussed in discussion section lines 434-437.
- The results about the immobilization of the enzyme into the magnetic particles are missing, what was the immobilization yield and recovered activity? Please add the results of these experiments.
We would like to thank the reviewer for his comment. This is present in lines 369-370.
Round 2
Reviewer 1 Report
The new manuscript version is better than the previous, however, the first comment of my previous review do not was addressed. The readers must understand clearly, why the authors speak about Mutagenesis, Enzyme immobilization and Magnetic nanoparticles in this work. At the beginning of each paragraph, the authors must explain, how and why these methods can improve enzymes or their productions. The authors must string together the ideas so that the objective of the work is clear at the end of the introduction. In the present manuscript the paragraphs: 3 (line 54), 4 (line 61) and 5 (line 67) just are ideas isolated.
Line 417 “The reactions surfaces analysis shows that the following factors: (aeration and peptone), (CaCl2 and peptone), (incubation period and peptone), (CaCl2 and aeration), (incubation period and aeration) are most influential to increase the reaction.”
The authors must be more precise in this conclusion. Why the factors are presented in pairs and repeated? They must give a precise prioritization of the factors that affect enzyme production based on the results of their experiments.
Minor corrections
In line 37 it is said “micro-organisms” instead of “microorganisms
Author Response
Response to reviewer 1 comments:
We would like to thank the reviewer for taking the time and effort necessary to review the manuscript. We sincerely appreciate all valuable comments and suggestions, which helped us to improve the quality of the manuscript.
Comments and Suggestions for Authors
The new manuscript version is better than the previous, however, the first comment of my previous review do not was addressed. The readers must understand clearly, why the authors speak about Mutagenesis, Enzyme immobilization and Magnetic nanoparticles in this work. At the beginning of each paragraph, the authors must explain, how and why these methods can improve enzymes or their productions. The authors must string together the ideas so that the objective of the work is clear at the end of the introduction. In the present manuscript the paragraphs: 3 (line 54), 4 (line 61) and 5 (line 67) just are ideas isolated.
The paragraph starting with line 54 was rewritten as per the reviewer comments.
The paragraph starting with line 61 was rewritten as per the reviewer comments.
The paragraph starting with line 67 was rewritten as per the reviewer comments.
Line 417 “The reactions surfaces analysis shows that the following factors: (aeration and peptone), (CaCl2 and peptone), (incubation period and peptone), (CaCl2 and aeration), (incubation period and aeration) are most influential to increase the reaction.”
The authors must be more precise in this conclusion. Why the factors are presented in pairs and repeated? They must give a precise prioritization of the factors that affect enzyme production based on the results of their experiments.
The lines describing the results of the previously deleted figure 3 is now deleted.
Minor corrections
In line 37 it is said “micro-organisms” instead of “microorganisms